# Concurrent Physiological and Pathological Angiogenesis in Retinopathy of Prematurity and Emerging Therapies

**DOI:** 10.3390/ijms22094809

**Published:** 2021-05-01

**Authors:** Chang Dai, Keith A. Webster, Amit Bhatt, Hong Tian, Guanfang Su, Wei Li

**Affiliations:** 1Department of Ophthalmology, Baylor College of Medicine, Houston, TX 77030, USA; chang.dai@bcm.edu (C.D.); keith.webster@bcm.edu (K.A.W.); arbhatt@texaschildrens.org (A.B.); 2Department of Ophthalmology, The Second Hospital of Jilin University, Changchun 130041, China; 3Department of Pharmacology, University of Miami School of Medicine, Miami, FL 33136, USA; 4Everglades Biopharma, LLC, Houston, TX 77030, USA; h.tian@evergladesbiopharma.com; 5Texas Children Hospital, Baylor College of Medicine, Houston, TX 77030, USA

**Keywords:** retinopathy of prematurity, oxygen-induced retinopathy, physiological angiogenesis, pathological angiogenesis, anti-angiogenic therapy, vascular endothelial growth factor, VEGF, secretogranin III, Scg3

## Abstract

Retinopathy of prematurity (ROP) is an ocular vascular disease affecting premature infants, characterized by pathological retinal neovascularization (RNV), dilated and tortuous retinal blood vessels, and retinal or vitreous hemorrhages that may lead to retinal detachment, vision impairment and blindness. Compared with other neovascular diseases, ROP is unique because of ongoing and concurrent physiological and pathological angiogenesis in the developing retina. While the disease is currently treated by laser or cryotherapy, anti-vascular endothelial growth factor (VEGF) agents have been extensively investigated but are not approved in the U.S. because of safety concerns that they negatively interfere with physiological angiogenesis of the developing retina. An ideal therapeutic strategy would selectively inhibit pathological but not physiological angiogenesis. Our group recently described a novel strategy that selectively and safely alleviates pathological RNV in animal models of ROP by targeting secretogranin III (Scg3), a disease-restricted angiogenic factor. The preclinical profile of anti-Scg3 therapy presents a high potential for next-generation disease-targeted anti-angiogenic therapy for the ROP indication. This review focuses on retinal vessel development in neonates, the pathogenesis of ROP and its underlying molecular mechanisms, including different animal models, and provides a summary of current and emerging therapies.

## 1. Introduction

Retinopathy of prematurity (ROP) with pathological retinal neovascularization (RNV) is the most common cause of blindness in children, primarily afflicting preterm infants [1]. During ROP, abnormal retinal blood vessels form sprouts and branches that extend not only over the retinal surface but also into the vitreous as they progress to pathological RNV and intravitreal neovascularization (IVNV) [2]. In contrast to other ocular neovascular diseases, including proliferative diabetic retinopathy (DR) with RNV and wet age-related macular degeneration (AMD) with choroidal neovascularization (CNV), ROP in the developing retina of premature infants is unique because of the coexistence of physiological and pathological angiogenesis. The former is essential for normal retinal development.

ROP first appeared as a clinical disorder in the 1940s, coincident with the liberal use of supplemental oxygen to improve the survival of preterm infants with respiratory distress syndrome (RDS) [3]. The disease typically occurs in preterm subjects that weigh about 1250 g or less and are born before 31 weeks of gestational age (GA). A full-term human pregnancy has a gestation period of ~40 weeks. Birth weight correlates inversely with the probability of ROP. There were ~14.9 million live premature births globally in 2010 [4], of which an estimated 184,700 developed ROP, including 20,000 with severe visual impairment and/or blindness [5]. Although about 90% of ROP infants sustain only mild visual impairment, subjects with untreated RNV remain at an elevated risk for progression to more severe conditions, including dilated and tortuous retinal vessels (i.e., the “plus” disease), retinal folds, pathological RNV and IVNV, and retinal or vitreous hemorrhage that may lead to retinal detachment, impaired vision and blindness (Figure 1).

ROP is currently treated by laser therapy or cryotherapy that provide only limited efficacy and significant risk of adverse side effects. Anti-vascular endothelial growth factor (VEGF) drugs have been used off-label in the U.S. but are not FDA approved because of the potential for adverse side effects caused by indiscriminate inhibition of all forms of angiogenesis in the developing retina. The recent discovery by our group that secretogranin III (Scg3) is a disease-selective angiogenic factor presents a unique opportunity to selectively target and alleviate ROP safely without affecting physiological angiogenesis and normal retinal development.

## 2. Physiological Angiogenesis in the Developing Retina

The retina is supplied by two vascular networks: the retinal vasculature supplies the inner retina while the choroidal vasculature supports the retinal pigment epithelium and photoreceptors (Figure 2) [6]. Abnormal growth of the choroidal vessels is responsible for wet AMD with CNV in the elderly, while ROP involves aberrant growth of the retinal vasculature in preterm neonates.

The human retina has two temporal vascular networks that develop sequentially in a precisely coordinated manner: the transient hyaloid vasculature supplies the retina and lens only in the early stage of gestation while the lifelong retinal vasculature supports only the retina. The hyaloid vasculature originates from the central hyaloid artery that enters the embryonic optic fissure at about 6 weeks of GA, progresses through the vitreous, surrounds the lens and then begins to regress about 13 weeks of GA [7].

Concomitantly, the human retinal vasculature begins to develop at about 16 weeks of GA in utero and is fully mature within 40 weeks of GA, normally before birth. Vascularization starts from the optic nerve head, extends toward the peripheral edge of the retina and forms the superficial retinal vascular plexus at about 36 weeks of GA. This vasculature further penetrates into the retina to form the intermediate and deep vascular plexuses. Together, these three layers of vessels form the retinal vascular network in the upper half of the retina, whereas the bottom half of the retina, including photoreceptors, remains avascular for the entire lifespan (Figure 2). The vessel-free photoreceptor layer, including the photoreceptor inner segments that contain a high density of mitochondria [8], is supplied by the retinal and choroidal vasculatures through diffusion.

Mice and rats share similar timelines of retinal vascularization, which are markedly different from humans. Mice are born with a hyaloid vasculature, and the retinal arteries sprout out of the optic nerve head into the retina on postnatal day 1 (P1), followed by outward branching and extension toward the peripheral retina [9]. Mouse superficial retinal vessels reach the edge of the retina by P8, penetrate into the retina from P7–12 and mature into superficial, intermediate and deep plexuses through vascular remodeling. Rat retinal vascular plexuses develop with similar timelines but are normally delayed by just a few days [10]. Importantly, in mice and rats, the retinal vasculatures develop postnatally, whereas in humans, they form prenatally.

Despite difference in timelines, humans and rodents share similar sequential patterns of hyaloid and retinal vasculature development. Additionally, both species have a network of astrocytes that spread outward from the optic nerve head before physiological RNV, and retinal vessels follow the network of astrocytes [9,11]. As oxygen tension in utero is relatively low, astrocytes under hypoxic conditions produce well-orchestrated gradients of angiogenic factors, including VEGF, insulin-like growth factor-1 (IGF-1) and erythropoietin (EPO) that support physiological angiogenesis in a stepwise manner directed toward the periphery to form a highly-structured retinal vascular network (Figure 2 and Figure 3B).

## 3. Two-Phase Theory of ROP with Pathological Angiogenesis

Supplemental oxygen triggers ROP in extremely low-birth weight preterm infants through two distinct phases (Figure 3A).

**Phase 1**: Vaso-obliteration is initiated at birth and subsides on termination of supplemental oxygen treatment, usually at approximately 32–34 weeks of GA. At birth, premature infants lose placental and maternal growth factors and are exposed to extrauterine high oxygen tension. Hyperoxia suppresses the expression of angiogenic factors that support physiological angiogenesis. Supplemental oxygen to treat RDS further downregulates angiogenic factors in the retina, leading to vaso-obliteration.

**Phase 2**: Vaso-proliferation begins at the time of withdrawal of supplemental oxygen. As preterm infants mature, the developing retina becomes metabolically active and relatively hypoxic because of delayed physiological RNV, causing upregulation of angiogenic factors, including VEGF and IGF-1 that trigger pathological RNV. Unlike the stepwise developmental program of physiological angiogenesis that is regulated by precise gradients of angiogenic factors diffusing from the astrocyte network, the burst of angiogenic factors during Phase 2 lacks such regulated temporal and spatial gradients, thereby triggering pathological RNV with a disorganized retinal vasculature (Figure 3B). Excessively expressed angiogenic factors may also accumulate in the vitreous, eliciting pathological IVNV.

## 4. Molecular Mechanisms of ROP

### 4.1. Oxygen

After low GA and birth weight, oxygen is the single most important risk factor for ROP. The liberal use of supplement oxygen in the 1940s resulted in a high incidence of ROP, first manifested as retrolental fibroplasia [3]. The subsequent development of technologies to better monitor and regulate oxygen exposure to premature infants decreased the risk of ROP, but reduction of supplemental oxygen also increased preterm infant mortality rates [12,13]. Clinical findings show that fluctuations of blood oxygen saturation (SpO_2_) of preterm babies during the first few weeks of life also contribute to the risk of ROP, and a high incidence of intermittent hypoxia is associated with severe ROP [14,15].

### 4.2. VEGF

VEGF is the most intensely studied angiogenic factor and widely considered to be the central player in both physiological and pathological angiogenesis of ROP as well as numerous other neovascular disorders. It was initially discovered as a vascular permeability factor and subsequently characterized as a potent mitogenic factor that stimulates endothelial proliferation and angiogenesis [16]. The mammalian VEGF family comprises five members, including VEGF-A, VEGF-B, VEGF-C, VEGF-D and placental growth factor (PlGF). Human VEGF-A undergoes alternative exon splicing into multiple variants, including VEGF111, VEGF121, VEGF145, VEGF165 and VEGF189 [16]. Of these isoforms, VEGF165 is the most active in terms of receptor stimulation and bioavailability. VEGF binds to 3 receptor tyrosine kinases, including VEGF receptor 1–3 (VEGFR1–3) encoded by *Flt1*, *Kdr* and *Flt4* genes, respectively [16]. VEGFR2 is the principal conduit that promotes endothelial proliferation and vascular permeability.

VEGF levels are significantly elevated in the vitreous of ROP infants [17,18]. Similarly, in mice with oxygen-induced retinopathy (OIR), a surrogate animal model of ROP, VEGF expression is suppressed by hyperoxia in Phase 1 but markedly upregulated by the conditions of relative hypoxia imposed in Phase 2 [19,20].

Oxygen tension regulates the expression of VEGF and multiple other angiogenic factors via hypoxia-inducible transcription factors (HIFs) [21]. HIF-1, the main regulator of responses to hypoxia consists of two subunits, HIF-1α and HIF-1β, both of which are required to activate target genes and are expressed constitutively. Oxygen tensions in the normal range (normoxia) or hyperoxia, drive prolyl hydroxylation of HIF-1α via prolyl hydroxylase domain proteins. Hydroxylated HIF-1α is sequestered to the proteasome for ubiquitination and degradation such that HIF-1 regulation is inactivated by normal (aerobic) or elevated oxygen pressure [21]. Under hypoxia, HIF-1α is stabilized and translocated to the nucleus where it heterodimerizes with HIF-1β, and the complex binds to hypoxia-responsive elements (HREs) of target genes, such as VEGF, to activate transcription. HIF-1 is induced during physiological angiogenesis as well as in Phase 2 of ROP but suppressed by supplemental oxygen during Phase 1 [7,22]. HIFs are central to both physiological and pathological angiogenesis in many organs [23].

### 4.3. IGF-1

IGF-1 is a non-oxygen regulated factor that has been extensively studied in ROP [24]. Clinical studies indicated that serum levels of IGF-1 in premature babies correlate inversely with the severity of clinical ROP [25]. Administration of recombinant human IGF-1 reduces the risk of OIR in mice [26]. Absence of IGF-1 in IGF-1^−/−^ mice delays normal retinal vascular development [25]. IGF-1 is a permissive factor for VEGF-dependent vascular endothelial cell growth [25,27,28].

IGF-1 binds the IGF-1 receptor (IGF-1R), and elimination of IGF-1R reduced RNV in OIR mice [29]. Insulin-like growth factor-binding proteins 1–6 (IGFBP1–6) bind and sequester >99% of available IGF-1, thereby regulating bioavailability. IGFBP-3, the most abundant IGFBP, sequesters 75–90% of IGF-1 protein [30,31]. IGFBP-3 concentrations are decreased in infants with ROP [32]. Deletion of the IGFBP-3 gene in mice resulted in a dose-dependent increase of OIR-related vessel loss, while administration of IGFBP-3 promoted vessel regrowth in wild-type mice and reduced vaso-obliteration in OIR mice [31,32].

### 4.4. Other Molecular Regulators

Erythropoietin, a key hematopoietic cytokine that regulates the formation of red blood cells during hematopoiesis, is also a pleiotropic growth factor that confers growth stimulation and cytoprotection to numerous cells, including endothelial cells. Chen et al. reported that EPO levels were suppressed in Phase 1 of OIR mice and markedly elevated in Phase 2 [33]. Exogenous EPO protected OIR retina in Phase 1 but worsened pathological RNV in Phase 2. EPO siRNA effectively suppressed pathological RNV in OIR mice [34]. Recombinant human EPO (rhEPO) used to treat anemia of prematurity is a significant independent risk factor for developing ROP [35].

ω-3 and ω-6 polyunsaturated fatty acids (PUFAs) are essential fatty acids required for optimal health and cannot be synthesized by the body. They are structural components of the plasma membrane and critical for optimal fluidity of photoreceptor membranes, retinal integrity and visual function [36]. ω-3 PUFAs include alpha-linolenic acid (ALA) and docosahexaenoic acid (DHA), while arachidonic acid (AA) is a ω-6 PUFA. DHA accounts for ~20% of total lipid in photoreceptor cells [37] and can be metabolized to neuroprotectin D1, resolvin D1 and resolvin E1, potent lipid-derived cytokines that confer protection against OIR-induced RNV [38]. Blockade of the peroxisome proliferator-activated receptor gamma (PPARγ) abrogated ω-3-mediated inhibition of pathological RNV through a VEGF-independent pathway [39], suggesting that ω-3 regulates angiogenesis through PPARγ. A clinical study revealed that low postnatal levels of serum AA are strongly associated with ROP development [40].

## 5. Animal Models of ROP

Animals with OIR have been widely used to model ROP and to investigate molecular mechanism and new therapies [41]. However, not all animals develop OIR. In some species, such as guinea pigs and rabbits, the retinal vasculature is absent or minimal, and the retina relies entirely on the diffusion from the choriocapillaris [42]. These animals cannot develop OIR. A critical prerequisite for animal OIR is a retinal vasculature at birth that mimics the human retina before 31 weeks of GA; namely, an immature retinal vasculature with active physiological angiogenesis. Animals with a fully developed retinal vasculature at birth, such as non-human primates (NHPs) and pigs [43], do not develop OIR, similar to the absent ROP in full-term human neonates. To date, OIR has been reported in only four mammalian species, including mice, rats, felines and dogs (Figure 4).

### 5.1. Mice

Mouse OIR is the most commonly used model to mimic RNV pathology of ROP. The retinal vascular maturity of mice at birth (P1) is equivalent to that of 25-week GA human fetuses [9], a time when the superficial vascular plexus starts to expand toward the peripheral retina.

The first mouse model of OIR was reported in 1994 by Smith et al. [44]. In this model, neonatal mice and their nursing mother were exposed to 75% oxygen from P7 to P12 (Figure 4A). After returning to room air, neonates initiated pathological RNV and IVNV at P14 that peaked at P17, followed by vaso-regression with almost complete resolution and replacement of aberrant vessels by a normal vasculature at P25.

The flat-mounted retina of OIR mice is characterized by peripheral aberrant RNV with a central avascular area (Figure 3B) [45,46]. Unlike the normal retinal vasculature with well-organized structural networks, the OIR vasculature is characterized by disorganized vascular networks with microaneurysm-like tufts (Figure 3B). Methods to quantify OIR neovascularization and neovascular tufts are well established [45]. Mouse OIR models may be influenced by a number of factors, including mouse strains, intra- and inter-litter variation, oxygen-related illness of lactating mothers, requirement of surrogate mothers, light exposure, effects of drug injection-related intraocular pressure on RNV, and quantification of OIR and data normalization.

### 5.2. Rats

Initial efforts to develop OIR in rats by exposing neonates to high oxygen for 5–10 days failed to reproduce ROP-like vascular manifestations [47]. Penn et al. successfully created a rat OIR model by exposing neonates to alternating high/low oxygen [48]. The success of this approach suggests that intermittent hyperoxia/hypoxia exacerbates OIR, consistent with clinical findings [14,15]. Subsequent studies indicated that exposure of neonatal rat pups to 50% oxygen at birth, followed by fluctuating oxygen cycles between 50% and 10% every 24 h for 14 days and finally, a 4-day post-exposure to room air provided optimal OIR in rats (Figure 4B) [48]. This model generates up to 97% incidence of pathological RNV with delayed physiological RNV and a peripheral avascular retina. Additionally, such an OIR retina has vascular tufts, dilated arteries and veins, high degrees of tortuosity of major arteries and frequent abnormal capillary buds. Severe intravitreal hemorrhage was also observed in about 42% of OIR rats. This model closely resembles ROP in humans.

### 5.3. Felines

Cats with a normal gestation of 63–67 days share a similar pattern of physiological RNV to humans and rodents but with an intermediate timeline. In kittens, the retinal artery emerges from the optical disc between the 35th and 45th days of GA but does not reach the periphery until 3 weeks after birth [49].

The first kitten model of OIR was established in 1954 by varying the onset, duration and concentrations of oxygen exposure [50]. Severe and even complete vaso-obliteration was achieved when kittens at P1–7 were exposed to 70–80% oxygen for 3–7 days and then returned to room air.

### 5.4. Dogs

Compared with rodent models, the canine retina is more vascularized at birth [51]. A dog model of OIR was developed by exposing 1-day-old purebred beagle neonates to 100% oxygen for 4 days (Figure 4C) [52]. On return to room air, the neonates displayed a period of marked pathological RNV that peaked between 3 and 10 days. At 22–45 days, OIR mimicked multiple ROP-like manifestations, including dilated and tortuous retinal vessels, mottled pigmentary changes, avascular regions in the peripheral retina, persistence of massive IVNV, tractional retinal folds, tented intravitreal vascularized membranes and vitreous synchysis [52,53]. Inner retinal astrogliosis was also detected by immunohistochemistry.

## 6. Current and Emerging Therapies

### 6.1. Oxygen for ROP Prevention

Over the past 70 years, numerous clinical trials have been implemented to investigate the optimal range of supplemental oxygen administered during Phase 1 and 2 to reduce the risk of ROP. These finding are summarized below. Detailed discussions on the clinical management of oxygen therapy are covered by other excellent reviews [1,54,55].

**Phase 1**: A clinical survey found that the risk and severity of ROP is associated with high SpO_2_ [56]. The rate of severe ROP was 5.5% for premature infants with a maximal SpO_2_ of >98% in the first two weeks of birth vs. 3% for those with maximal SpO2 of ≤98%. After 2 weeks of age, the rate of severe ROP was 3.3% with maximal SpO_2_ of >92% vs. 1.3% with SpO2 of ≤92%.

Five large multicenter, randomized, masked, controlled trials, collectively known as The Neonatal Oxygen Prospective Meta-Analysis or NeOProM Collaboration, enrolled approximately 5000 premature infants <28 weeks, including the SUPPORT trial in the U.S., BOOST II trials in Australia, New Zealand and United Kingdom, and the COT trial in Canada. Compared with premature infants exposed to high SpO_2_ (91–95%), infants with low SpO_2_ (85–89%) sustained a marked reduction in severe ROP, but again, the benefit was accompanied by significantly increased mortality [13,57].

**Phase 2**: Based on the two-phase theory of ROP pathogenesis, it was postulated that increased oxygen supplementation in Phase 2 (typically after 32 weeks of GA) would mitigate the disease. A multicenter STOP-ROP clinical trial found that high oxygenation (SpO_2_ 96–99%) neither caused additional progression of pre-threshold ROP nor significantly reduced the number of infants requiring peripheral ablative surgery [58]. This finding is consistent with the Australian BOOST trial, in which ROP infants of less than 30 weeks of GA were treated with an SpO_2_ range of either 91–94% or 95–98% at 32 weeks of GA [59]. Again, targeting a higher oxygen-saturation range in Phase 2 conferred no significant benefit.

In summary, numerous large multicenter clinical trials implemented over the previous 70 years were unable to identify an optimal oxygen saturation range that favorably balanced the risks between mortality and ROP in Phase 1, and no clinical benefit was demonstrated for high oxygen treatment in Phase 2. Despite the setbacks, valuable knowledge generated from these studies has improved clinical management of ROP. Firstly, exposure of premature infants to excessively high levels of supplemental oxygen confers increased risk of ROP. Optimal SpO_2_ is within the range of 90–95% and should be personalized within this range as indicated for individual cases [60]. Second, fluctuations in SpO_2_ during the first few weeks of life are associated with increased risk and severity of ROP. Therefore, strict management to avoid fluctuation in SpO2 is important. Additional clinical trials are warranted to optimize oxygen therapy and reduce ROP.

### 6.2. Laser Therapy and Cryotherapy

Cryotherapy and laser photocoagulation are ablative surgeries to destroy non-neovascularized area in the peripheral retina and convert them into non-functional scar tissues for preserving the central vision and are reviewed in detail elsewhere [1,61]. In brief, both treatments reduce unfavorable outcomes compared with untreated control eyes but with limited efficacy and potential adverse side effects, including loss of peripheral vision, scar induction, inflammation and myopia [61,62]. Laser photocoagulation has become the current choice for ROP treatment because of its convenient administration through the anterior of the eye, decreased requirement for general anesthesia and relatively low rate of systemic complications. Neither therapy addresses the underlying mechanisms of pathological RNV in ROP.

### 6.3. Anti-VEGF Therapy

Given the critical role of VEGF in pathological RNV of ROP and the clinical successes of anti-VEGF agents for other neovascular indications, anti-VEGF has generated interests as a therapeutic option for ROP. However, safety is a major concern for anti-VEGF therapy of ROP. VEGF is an essential growth factor for physiological angiogenesis that supports the development of the retina and other organs during embryonic and neonatal stages. Indeed, mice with homozygous deletion of either VEGFR1 or 2 die in utero [63,64]. Similarly, mice with deletion of a single VEGF allele are embryonic lethal [65]. Embryos of all mice with the deletion of VEGF or VEGFR exhibit severe defects in vasculogenesis and embryogenesis.

Currently, there are five approved anti-VEGF drugs for wet AMD and diabetic retinopathy, including ranibizumab, pegaptanib, aflibercept, brolucizumab and conbercept (approved in China). Additionally, bevacizumab approved for cancer therapy is often used off-label for ocular diseases [66].

Bevacizumab was the first anti-VEGF drug to be reported for ROP therapy [67]. BEAT-ROP multicenter, randomized, controlled trial compared intravitreal injection of bevacizumab with laser therapy [68]. The results revealed an advantage of intravitreal bevacizumab over laser therapy for Stage 3+ ROP infants with Zone I but not Zone II disease. However, the trial with only 143 included infants was too small to assess safety. Significant vascular and macular abnormalities in ROP eyes treated with intravitreal bevacizumab were described in other clinical trials [69,70,71]. Anti-VEGF therapy in additional clinical studies and case reports of ROP were associated with multiple serious adverse outcomes, including retinal hemorrhage and detachment [72,73,74,75]. Furthermore, ROP recurrence after intravitreal bevacizumab is not uncommon [76]. Such results have presented setbacks for the continued development of bevacizumab as a therapy for ROP.

The randomized, multicenter RAINBOW trial recently found that intravitreal ranibizumab (0.2 mg/eye) was superior to laser therapy with fewer unfavorable structural outcomes and an acceptable 24-week safety profile [77]. Based on this trial, ranibizumab was approved for ROP therapy in the European Union in 2019.

In ROP infants, intravitreal injection of bevacizumab or ranibizumab resulted in significantly decreased serum VEGF up to 12 weeks [78]. Given the neurotrophic actions of VEGF [79], circulating anti-VEGF drugs caused by leakage from the eye may adversely affect brain development in anti-VEGF-treated ROP infants. Indeed, clinical studies found that ROP infants treated with bevacizumab demonstrated lower motor scores and higher rates of severe neurodevelopmental disability in comparison with laser therapy at 18 months of age [80,81].

We recently reported that intravitreal aflibercept inhibited vessel development in the retina, leading to reduced retinal vascularization and decreased b-wave amplitude of electroretinography (ERG) [82]. Systemic aflibercept also adversely affected the kidney with reduced vascular density, dilated tubules, abnormal glomeruli and retarded body weight gain. A separate study reported that intravitreal aflibercept disrupted the retinal architecture and reduced ERG b-wave amplitude in OIR mice [83]. A previous study revealed that intravitreal aflibercept inhibited development of the superficial and deep retinal plexus in healthy neonatal dogs with a narrow therapeutic window for OIR [52]. These findings confirm in animal models that VEGF inhibition not only suppresses pathological RNV/IVNV but also physiological angiogenesis. Depletion of VEGF-dependent neurotrophic actions on retinal neurons and disruption of physiological angiogenesis are the most likely causes of the observed multi-organ adverse side effects. Therefore, because of the evidence for ROP recurrence and continued safety concerns as well as the disputed risk-benefit profiles, further tests of anti-VEGF therapy for ROP will require careful monitoring of both safety and efficacy to determine whether there is a place for such agents in the treatment of ROP preterm infants.

### 6.4. Other Therapies

Over the past 70 years, numerous clinical trials have failed to identify effective and safe therapeutic strategies for ROP [55,84]. Such trials have included applications of recombinant human IGF-1 (rhIGF-1) in combination with rhIGFBP-3 [85], anti-oxidant vitamin E [86,87], inositol [88] and ω-3 PUFAs [89].

Oral propranolol, a non-selective β-blocker that has been extensively used in newborn infants with indications for other disorders, effectively reduced severe ROP but with serious adverse side effects [90,91]. Clinical trials further showed that propranolol delivered by topical eye drop was well tolerated and also effectively suppressed the progression of ROP to advanced stages at a dose of 0.2% but not 0.1% [92]. Based on these findings, additional clinical studies of propranolol with larger sample sizes, appropriate dose ranges and administration routes are warranted.

### 6.5. Scg3 Antagonist as An Emerging Therapy That Targets Pathological Angiogenesis

By applying a new technology called comparative ligandomics, our group recently discovered that Scg3 is a novel disease-restricted pro-angiogenic factor [46]. Of thousands of endothelial ligands identified by the technology, Scg3 had the highest binding activity ratio (1731:0) to diabetic vs. healthy retinal vessels; whereas VEGF binds equally well to both diabetic and healthy retinal vasculatures. In vivo functional assays confirmed that Scg3 selectively promotes angiogenesis and retinal vascular leakage in diabetic but not healthy mice [46,93]. In contrast, VEGF induces angiogenesis and retinal vascular permeability in both diabetic and healthy mice.

These findings suggest that an as-yet-unknown Scg3 receptor (Scg3R) is markedly upregulated on the surface of diabetic endothelial cells (by >1700-fold) to account for the enhanced binding with only moderately increased ligand Scg3. We found only a 1.38-fold increase of Scg3 in diabetic vitreous fluid versus controls [46]. This is in contrast to the marked 36-110-fold upregulation of VEGF in proliferative diabetic retinopathy with a minimal 2.5-fold increase of the VEGFR1 [94,95]. Based on these data, we proposed that angiogenic factors contribute to pathogenic angiogenesis through two different modes of action: upregulation of ligands or their cognate receptors on endothelial cells with comparable outcomes. The two modes have distinct implications for targeted anti-angiogenic therapy. Induced secretory ligands migrate extracellularly and regulate both diseased and healthy endothelial cells, whereas increased angiogenic receptors on the surface of cells restricted to a diseased environment are stationary and only responsive within such a disease state. Therefore, anti-angiogenic therapy against unrestrictive ligands, such as VEGF, disrupts vascular growth equally in both diseased and healthy conditions. In contrast, therapies that target disease-selective factors, such as Scg3, are predicted to selectively block pathological angiogenesis without affecting the growth of healthy vessels.

To investigate the novel concept of ligand-guided disease-targeted anti-angiogenic therapy, we developed Scg3-neutralizing monoclonal antibodies (mAbs) and confirmed their high efficacy to alleviate retinal vascular leakage in diabetic mice [46]. The results suggest that anti-Scg3 mAbs are effective reagents to treat DR in mouse models and represent candidates for clinical development to treat patients with diabetic macular edema. We also found that anti-Scg3, like anti-VEGF, has multiple indications to alleviate ocular neovascular disorders in mouse models, including OIR and CNV [46,82,96]. Compared with aflibercept that elicits side effects in neonatal mice with developing retinal vessels, we confirmed that anti-Scg3 mAb, administered either intravitreally or systemically, blocked pathological angiogenesis with no detectable side effects on healthy vasculatures [82]. The results strongly support the concept of next-generation anti-angiogenic therapy to selectively target pathological angiogenesis. Reports that Scg3^−/−^ mice have a normal gross phenotype attest to the predicted safety of anti-Scg3 therapy in OIR mice and possibly human ROP by extension [97]. In contrast, VEGF^−/+^ mice are embryonic lethal with severe defects in vasculogenesis [65]. A caveat is that Scg3 disease selectivity has been determined and quantified by comparative ligandomics [46] but is yet to be experimentally confirmed for OIR and ROP.

## 7. Perspectives

Due to the prerequisite to selectively target pathological angiogenesis while sparing physiological angiogenesis, ROP represents a unique clinical challenge (Figure 5). This contrasts with many other clinical conditions of pathological angiogenesis in adults that can tolerate pan-angiogenic inhibition, at least in the short term, because of mature and often aged vasculatures that are not so reliant on active physiological angiogenesis. Such conditions are perhaps best represented by adult-onset ocular vascular diseases and oncogenic tumors that respond positively to anti-VEGF therapy without major adverse side effects. Treatment of human ROP infants must be compatible with normal ocular, neurological and systemic physiological growth and development. To date, no therapeutic strategy can achieve these criteria, and ROP remains an urgent unmet clinical need with little promise from current pipeline candidates.

Ligand-guided disease-targeted anti-Scg3 therapy offers a new hope to selectively suppress pathological angiogenesis with minimal inhibition of physiological angiogenesis and a high safety profile. This is in marked contrast with anti-VEGF-based therapies that lack the obligatory vascular selectivity to treat human ROP. Despite our reports of safety and efficacy in a mouse OIR model [82], the therapeutic windows of anti-Scg3 still need to be precisely determined and compared with anti-VEGF in mice as well as non-murine preclinical models before we can advance the technology to our final goal of anti-Scg3 clinical trials for ROP.

## Figures and Tables

**Figure 1 ijms-22-04809-f001:**
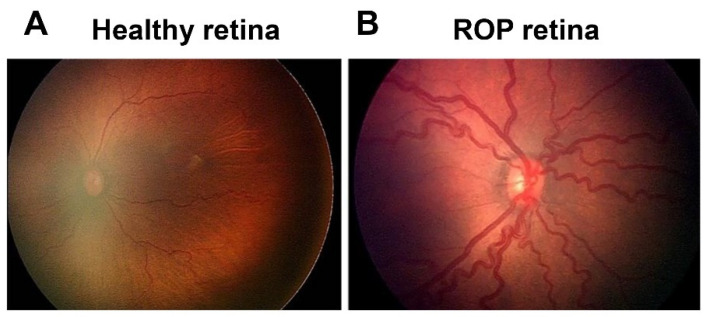
Fundus image of retinopathy of prematurity (ROP): (**A**) Healthy retina and (**B**) ROP retina with tortuous arteries and dilated veins as plus disease.

**Figure 2 ijms-22-04809-f002:**
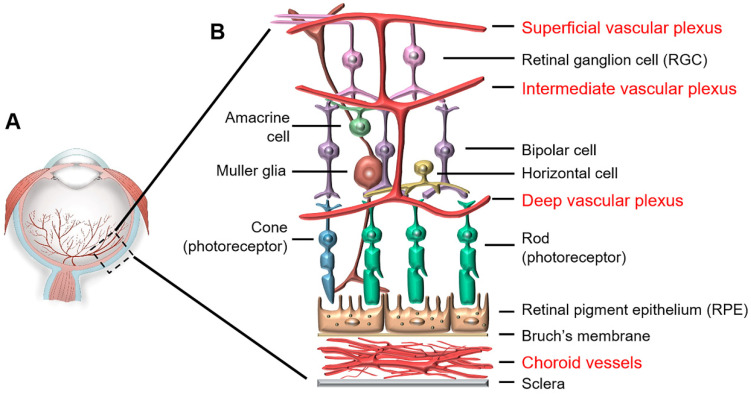
Retinal structure and vascular systems: (**A**) Eye structure and (**B**) Retinal structure and vascular systems. The retina is supplied by two vascular systems: retinal and choroidal vasculatures. The retinal vasculature comprises three layers, including the superficial, intermediate and deep plexuses that are interconnected and supply the upper half of the retina. The bottom half of the retina is avascular, relying on the diffusion from the retinal and choroidal vessels.

**Figure 3 ijms-22-04809-f003:**
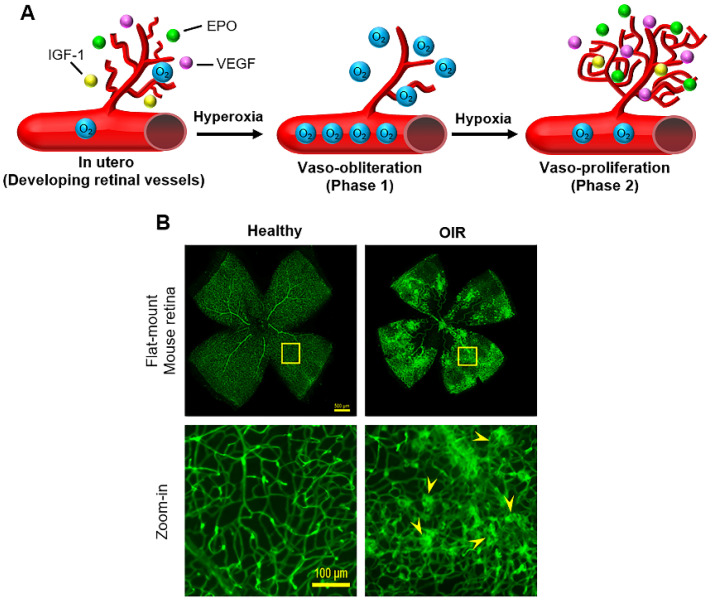
(**A**) Two phases of human ROP and (**B**) Healthy and OIR flat-mount mouse retina stained with Alexa Fluor 488-conjugated isolectin B4. Scale bar = 500 (top row) and 100 μm (bottom row). Arrowheads indicate pathological retinal neovascularization and neovascular tufts.

**Figure 4 ijms-22-04809-f004:**
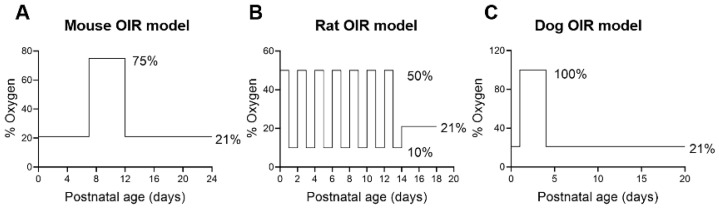
Animal OIR models: (**A**) Mouse model of OIR; (**B**) Rat OIR model and (**C**) Dog model of OIR. There is no consensus on an optimal feline OIR model.

**Figure 5 ijms-22-04809-f005:**
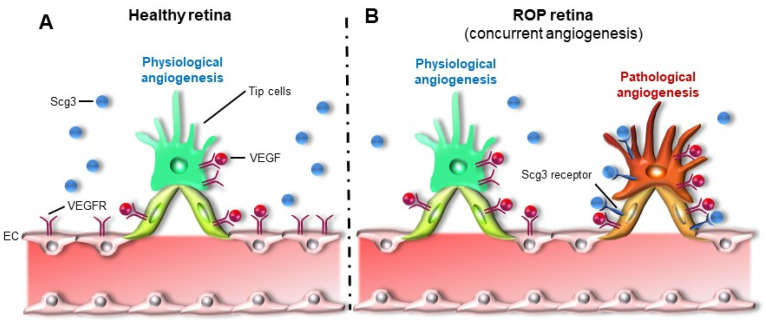
Concurrent physiological and pathological angiogenesis in ROP: (**A**) Physiological angiogenesis in the healthy retina and (**B**) Concurrent physiological and pathological angiogenesis in the ROP retina. Scg3 receptor is upregulated on pathological neovessels with minimal induction of Scg3 ligand itself, as in contrast to induction of VEGF ligand in ROP.

## Data Availability

Not applicable.

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
