# Peer review of "Concurrent Physiological and Pathological Angiogenesis in Retinopathy of Prematurity and Emerging Therapies"

_ijms, 2021, doi:10.3390/ijms22094809_

Round 1
Reviewer 1 Report
The review is well conceptualized. The title and conclusion section (perspectives) are suitable. The citations are current, and the findings in the field are well presented.
If addressed, the comment below could improve the manuscript friendliness. The text describing Figure 3 (lines 117-135) does not mention Figure 3 B, D, E until Section 5 (lines 228), Section 5.2 (line 256), section 5.4 (line 275). Presenting the data describing OIR of Figure 3 before the figure would be helpful.
Author Response
Response: To address issue, we rearranged original Figure 3 into new Figure 3 and 4 (see our response to Reviewer 2, Comment #1). New Figure 3A is mentioned on Line 132, and new Figure 3B is listed on Line 146.
Line 128-129: Change “(Figure 2, 3C)” to “(Figure 2, 3B)”.
Line 249: Change “(Figure 3B-E)” to “(Figure 4)”.
Line 256-257: Change “(Figure 3B)” to “(Figure 4A)”
Line 261: Change “(Figure 3C)” to “(Figure 3B)”
Line 263: Change “(Figure 3C)” to “(Figure 3B)”
Line 277: Change “(Figure 3D)” to “(Figure 4B)”
Line 296: Change “(Figure 3E)” to “(Figure 4C)”
Line 460: Change “(Figure 4)” to “(Figure 5)”
Line 480: Change “Figure 4” to “Figure 5”
Reviewer 2 Report
An excellent review and beautifully written !
Only few suggestions:
1) figures should be inserted into the main text closer to their first citation, this was not applied for Figure 3B,D, and E. The authors can re-organise the panels in Figure 3 so that panels B,D,E are moved into a new figure and inserted under the appropriate section "5.Animal Models of ROP" while Figure 3 would contain only panels A and C
2) The feline model of OIR is not represented with a chart explaining the strategy like the other models, was this an unintentional oversight or there is a reason?
3) line 180-181: the statement is missing a supporting citation
Author Response
Reviewer 2
1) figures should be inserted into the main text closer to their first citation, this was not applied for Figure 3B,D, and E. The authors can re-organise the panels in Figure 3 so that panels B,D,E are moved into a new figure and inserted under the appropriate section "5.Animal Models of ROP" while Figure 3 would contain only panels A and C.
Response: We reorganized original Figure 3 by grouping original Figure 3A and C as new Figure 3A and B and original Figure 3 B, D and E as new Figure 4 A, B and C. We also split the legend of original Figure 3 into legends of new Figure 3 (use the attached new figures to replace old Figure 3 at Line 148-150) and Figure 4 (Line 234-236).
Line 151-156: Replace the figure legend of Figure 3 with the following:
Figure 3. (A) Two phases of human ROP. (B) Healthy and OIR flat-mount mouse retina stained with Alexa Fluor 488-conjugated isolectin B4. Scale bar = 500 (top row) and 100 μm (bottom row). Arrowheads indicate pathological retinal neovascularization and neovascular tufts.
Line 237-238: Add the following legend of new Figure 4:
Figure 4. Animal OIR models. (A) Mouse model of OIR. (B) Rat OIR model. (C) Dog model of OIR. There is no consensus feline model of OIR.
2) The feline model of OIR is not represented with a chart explaining the strategy like the other models, was this an unintentional oversight or there is a reason?
Response: We purposely skipped the chart for feline model of OIR in new Figure 4 because oxygen concentration, duration of exposure and kitten ages varied in different studies. This disease model was less frequently used. In the legend of new Figure 4, we state that “There is no consensus on an optimal feline OIR model.”
3) line 180-181: the statement is missing a supporting citation.
Response: Line 197: We inserted a reference, “[23]”.
Additionally, we made the following revisions.
1) Line 13: Add “wei.li4@bcm.edu”
2) Line 85-90: Add missing paragraph.
3) Line 94-101: Replace incorrect the legend of Figure 2.
4) High-resolution images (>600 dpi) of Figure 1 -5 in TIFF format are attached.
Line 57-66: Replace Figure 1 image with the attached high-resolution image of Figure 1.
Line 91-93: Replace Figure 2 image with the attached high-resolution image of Figure 2.
Line 148-150: Replace Figure 3 image with the attached high-resolution image of the revised Figure 3.
Line 234-236: Add the attached high-resolution image of new Figure 4.
Line 307: Change “discussion” to “discussions”
Line 369: Change the font of “I”
Line 391-392: Change “retarded body weight gain, dilated tubules and abnormal glomeruli” to “dilated tubules, abnormal glomeruli and retarded body weight gain”.
Line 393: change “b-amplitude” to “b-wave amplitude”.
Line 477-479: Replace Figure 5 image with the attached high-resolution image of Figure 5.